# Mitochondrial Transport Proteins in Cardiovascular Diseases: Metabolic Gatekeepers, Pathogenic Mediators and Therapeutic Targets

**DOI:** 10.3390/ijms26178475

**Published:** 2025-08-31

**Authors:** Yue Pei, Sitong Wan, Jingyi Qi, Xueyao Xi, Yinhua Zhu, Peng An, Junjie Luo, Yongting Luo

**Affiliations:** 1Department of Nutrition and Health, China Agricultural University, Beijing 100193, China; 2023333010129@cau.edu.cn (Y.P.); adeline7wan@cau.edu.cn (S.W.); qipeiyan2992@163.com (J.Q.); xiyaoyao0621@163.com (X.X.); zhuyinhua@cau.edu.cn (Y.Z.); an-peng@cau.edu.cn (P.A.); 2College of Biological Sciences, China Agricultural University, Beijing 100193, China

**Keywords:** mitochondrial transport proteins, cardiovascular diseases (CVDs), metabolic reprogramming, mitochondrial dysfunction, therapeutic targets

## Abstract

Mitochondria, as the metabolic hubs of cells, play a pivotal role in maintaining cardiovascular homeostasis through dynamic regulation of energy metabolism, redox balance, and calcium signaling. Cardiovascular diseases (CVDs), including heart failure, ischemic heart disease, cardiomyopathies, and myocardial infarction, remain the leading cause of global mortality, with mitochondrial dysfunction emerging as a unifying pathological mechanism across these conditions. Emerging evidence suggests that impaired mitochondrial transport systems—critical gatekeepers of metabolite flux, ion exchange, and organelle communication—drive disease progression by disrupting bioenergetic efficiency and exacerbating oxidative stress. This review synthesizes current knowledge on mitochondrial transport proteins, such as the voltage-dependent anion channels, transient receptor potential channels, mitochondrial calcium uniporter, and adenine nucleotide translocator, focusing on their structural–functional relationships and dysregulation in CVD pathogenesis. We highlight how aberrant activity of these transporters contributes to hallmark features of cardiac pathology, including metabolic inflexibility, mitochondrial permeability transition pore destabilization, and programmed cell death. Furthermore, we critically evaluate preclinical advances in targeting mitochondrial transport systems through pharmacological modulation, gene editing, and nanoparticle-based delivery strategies. By elucidating the mechanistic interplay between transport protein dysfunction and cardiac metabolic reprogramming, we address a critical knowledge gap in cardiovascular biology and provide a roadmap for developing precision therapies. Our insights underscore the translational potential of mitochondrial transport machinery as both diagnostic biomarkers and therapeutic targets, offering new avenues to combat the growing burden of CVDs in aging populations.

## 1. Introduction

Cardiovascular diseases (CVDs)—encompassing heart failure, ischemic heart disease, cardiomyopathies, and myocardial infarction—constitute the foremost cause of global mortality [1]. Alarmingly, the prevalence of CVD subtypes such as ischemic heart disease and heart failure continues to escalate, driven by aging populations and metabolic epidemics. Recent epidemiological analyses reveal that ischemic heart disease-associated heart failure alone afflicted over 19.16 million individuals globally in 2021 [2]. This rising burden underscores an urgent need to redefine pathogenic mechanisms and therapeutic strategies beyond conventional approaches.

Central to this pursuit is the mitochondrion, the metabolic nexus of cells, whose dysfunction acts as a convergence point for CVD pathogenesis. Mitochondrial failure manifests through bioenergetic collapse, redox imbalance, and calcium mishandling—processes amplified by mitochondrial DNA (mtDNA) mutations, oxidative phosphorylation (OXPHOS) defects, and defective quality control mechanisms [3,4]. Additionally, ferroptosis—an iron-dependent form of regulated cell death driven by lipid peroxidation—has emerged as a critical contributor to cardiovascular pathogenesis [5]. Excessive reactive oxygen species (ROS) production not only directly damages macromolecules but also triggers maladaptive inflammatory cascades, accelerating endothelial dysfunction, atherosclerotic plaque instability, and post-infarction remodeling [6,7,8,9]. Compromised mitophagy further exacerbates this cycle, allowing accumulation of ROS-generating damaged mitochondria in conditions like ischemia–reperfusion injury and diabetic cardiomyopathy [7]. Simultaneously, disrupted mitochondrial dynamics—excessive fission and suppressed fusion—destabilize cardiac energy metabolism, precipitating apoptosis and contractile dysfunction [10,11,12]. These interrelated pathways position mitochondrial integrity as both a biomarker and a modifiable axis in CVD progression.

Emerging evidence now implicates mitochondrial transport proteins—gatekeepers of metabolite flux and organelle communication—as master regulators of this pathophysiological cascade [13]. These specialized transporters, including the voltage-dependent anion channels (VDACs), mitochondrial calcium uniporter (MCU), and adenine nucleotide translocator (ANT), orchestrate substrate shuttling, ion homeostasis, and redox signaling across mitochondrial membranes [14]. Their dysfunction rewires cardiac metabolism: for instance, endothelial VDAC downregulation in atherosclerosis shifts fatty acid utilization toward lipotoxicity, while MCU hyperactivity in hypertension drives calcium overload-induced vascular hypertrophy [15,16]. Such isoform-specific dysregulation creates distinct metabolic fingerprints in CVD subtypes, linking transport protein aberrations to disease-specific phenotypes like diastolic stiffness or plaque rupture. Critically, these proteins act as nodal points integrating environmental stressors (e.g., hyperglycemia, hypertension) with mitochondrial maladaptation, offering mechanistic insights into the heterogeneity of CVD manifestations.

This review summarizes cutting-edge advances in mitochondrial transport proteins as dynamic arbiters of cardiovascular health and disease. We first delineate the structural and functional diversity of these transporters, emphasizing their roles in metabolic flexibility and stress adaptation. Next, we dissect their isoform-specific pathogenic contributions across CVD spectra, from initiating metabolic inflexibility to executing terminal cell death pathways. Finally, we critically evaluate innovative targeting strategies—including allosteric modulators, mitochondrial-targeted nanoparticles, and CRISPR-based precision editing—that are reshaping therapeutic paradigms. By bridging molecular insights with translational opportunities, this work aims to catalyze the development of mechanistically grounded, personalized therapies for CVDs rooted in mitochondrial transport biology.

## 2. Classification of Mitochondrial Transporters Potentially Regulating CVDs

Mitochondrial transport proteins act as critical metabolic gatekeepers, regulating ion/metabolite flux to sustain bioenergetics and redox balance. Their dysfunction drives cardiovascular pathogenesis through disrupted calcium handling, oxidative stress, and metabolic reprogramming. This section classifies key transporters—including porins (VDACs), ion channels (TRP/MCU), and solute carriers (SLC25)—by structure, localization, and physiological roles. We highlight isoform-specific pathogenic mechanisms in CVDs, supported by Table 1’s systematic framework of their disease implications.

### 2.1. Porin Channels

#### Voltage-Dependent Anion Channels

The VDAC, a quintessential β-barrel protein of the outer mitochondrial membrane (OMM), governs critical cellular processes including transmembrane transport of ATP/ADP, maintenance of calcium (Ca^2+^) homeostasis, regulation of Ca^2+^ flux at endoplasmic reticulum-mitochondria junctions, and modulation of apoptotic signaling cascades [28,29]. As one of the most abundantly expressed proteins in the OMM [30], mammalian VDACs comprise three genetically distinct isoforms—VDAC1, VDAC2, and VDAC3. VDAC1, the most extensively characterized isoform, serves as a molecular hub linking metabolic dysfunction to cell death through interactions with Bcl-2 family proteins. VDAC1 is a key mediator of mitochondria-driven apoptosis, while VDAC2, conversely, serves an anti-apoptotic function [31]. VDAC2 is involved in the regulation of calcium signaling in cardiac cells. It interacts with ryanodine receptor 2 to facilitate calcium transport from the sarcoplasmic reticulum to the mitochondria, which is essential for the proper functioning of cardiac myocytes. Notably, VDAC2 repair restores contractile function in failing hearts by rescuing mitochondrial–calcium coupling. VDAC2-targeted compound reverses pathological hypertrophy in preclinical models [32]. VDAC3 imports glutathione (GSH) to neutralize mitochondrial ROS, protecting against oxidative damage in diabetic cardiomyopathy [33,34]. Its loss disrupts fatty acid transport into mitochondria, reducing β-oxidation efficiency and promoting lipotoxicity in obese cardiomyopathy models [35]. VDAC3 also modulates mitochondrial dynamics via interactions with fission/fusion proteins, linking organelle morphology to cardiac energetics [29].

### 2.2. Ion Channels

#### 2.2.1. Transient Receptor Potential (TRP) Channels

The TRP ion channel family is a diverse group of proteins. Based on sequence and topological differences, mammalian TRPs can be categorized into six subfamilies: TRPC, TRPV, TRPM, TRPA, TRPP, and TRPML [36]. These channels are characterized by six transmembrane spanning domains (S1–S6), with a pore-forming loop between S5 and S6 [37].

##### TRPC Subfamily

TRPC, as nonselective cation channels, have a broad tissue-specific distribution and are involved in various pathophysiological functions [38]. Its dysfunction may lead to vascular constriction, smooth muscle cell proliferation, and vascular remodeling, promoting the development of vascular diseases such as hypertension and atherosclerosis [39,40]. TRPC6 regulates Ca^2+^ influx and ROS production, TRPC1 forms heteromeric channels for Ca^2+^-dependent signaling [41,42].

##### TRPV Subfamily

The TRPV ion channel family, which includes TRPV1-TRPV4 and TRPV5-TRPV6 [36]. Activation of TRPV1 permits the influx of Ca^2+^ and Na^+^ into cells, thereby triggering a multitude of signaling pathways involved in various physiological and pathological processes. Studies have shown that activation of TRPV1 by capsaicin can increase the formation of mitochondria-associated endoplasmic reticulum membranes, which are essential for mitochondrial calcium transport, oxidative capacity, and metabolism. This protective effect is mediated through the AMPK-MFN2 signaling pathway [43]. TRPV4 activation in vascular smooth muscle cells increases mitochondrial ROS via Ca^2+^-dependent RhoA/ROCK signaling, promoting vascular stiffness and hypertension [44]. Inhibition of TRPV4 restores endothelial nitric oxide (NO) bioavailability, improving metabolic flexibility [45].

#### 2.2.2. Calcium Transporters

Calcium transporters are essential proteins that regulate intracellular Ca^2+^ homeostasis, including calcium channels, sodium–calcium exchangers (NCX), and the mitochondrial calcium uniporter (MCU). Calcium channels allow Ca^2+^ influx into cells, which is vital for processes like excitation-contraction coupling in cardiomyocytes [46]. Sodium–calcium exchangers, such as NCX1, expel Ca^2+^ from cells to maintain low intracellular Ca^2+^ levels, crucial for preventing calcium overload [47,48].

The MCU complex is the primary pathway for Ca^2+^ entry into mitochondria and is located in the inner mitochondrial membrane (IMM). This complex is critical for mitochondrial Ca^2+^ homeostasis and cellular energy metabolism. The MCU complex consists of several key subunits: MCU (the pore-forming subunit), EMRE (essential for MCU assembly and stability), MICU1/2 (regulatory subunits that act as Ca^2+^ sensors, inhibiting MCU activity at low cytosolic Ca^2+^ concentrations [<1 μM] and activating it at higher levels [>2 μM]) [49], and MCUb. MCUb forms low-activity heteromeric channels with MCU, reducing mitochondrial Ca^2+^ uptake conductance.

#### 2.2.3. Potassium/Sodium/Magnesium Transporters

##### K^+^ Channels

Mitochondrial K^+^ channels are integral membrane proteins located in the IMM that facilitate the transport of potassium ions across the membrane. These channels are essential for maintaining mitochondrial membrane potential (ΔΨm) and regulating mitochondrial function. They are activated by changes in intracellular K^+^ concentration, membrane potential, and other stimuli such as pH and Ca^2+^ levels [50]. Mitochondrial Big-Conductance Ca^2+^-Activated K^+^ Channels (mitoBKCa) are a specialized subset of BKCa channels that are localized to the IMM. These channels are characterized by their large conductance and sensitivity to intracellular Ca^2+^ levels. They play a crucial role in regulating ΔΨm and intracellular Ca^2+^ homeostasis [51]. The activation of mitoBKCa channels allows K^+^ ions to flow out of the mitochondria, which helps stabilize the mitochondrial membrane potential and prevent excessive Ca^2+^ accumulation. The primary function of mitoBKCa channels is to modulate mitochondrial membrane potential and Ca^2+^ homeostasis [52]. Overexpression or altered activity of mitoBKCa channels can lead to mitochondrial Ca^2+^ overload, impaired OXPHOS, and increased oxidative stress, all of which contribute to cellular dysfunction and energy deficits [53]. mitoBKCa overactivation in diabetic cardiomyopathy induces pathological K^+^ efflux, dissipating ΔΨm and causing mitochondrial calcium overload, oxidative stress, and impaired ATP synthesis that accelerates cardiac dysfunction, while targeted inhibition reduces infarct size in ischemia–reperfusion models by preventing ΔΨm collapse. These changes can further impair cardiac function and contribute to the progression of heart failure [54].

##### Voltage-Gated Na^+^ Channels

Na^+^ channels are integral membrane proteins that play a crucial role in the initiation and propagation of action potentials in excitable cells such as neurons and muscle cells, including cardiomyocytes. Voltage-gated sodium channels (Nav) are critical for cardiac action potential initiation and propagation, primarily mediated by the Nav1.5 subtype in cardiomyocytes. Dysfunctional Nav1.5 activity prolongs action potential duration, induces intracellular sodium overload, and disrupts calcium homeostasis via Na^+^/Ca^2+^ exchanger (NCX) activation, leading to delayed afterdepolarizations and ventricular arrhythmias in heart failure [55]. The mitochondrial Sodium/Proton Exchanger (NHE) is a crucial protein located in the inner mitochondrial membrane that facilitates the exchange of Na^+^ for protons (H^+^) across the mitochondrial membrane [56]. The primary function of the mitochondrial NHE is to regulate the mitochondrial membrane potential and intracellular pH by exchanging Na^+^ for H^+^ [57]. The mitochondrial NHE is involved in the regulation of intracellular Ca^2+^ homeostasis. During ischemia, the balance of Ca^2+^ is disrupted, leading to ROS generation and mitochondrial dysfunction. The activation of the NHE helps counteract these effects by maintaining pH balance and preventing excessive acidification [58]. Epithelial Sodium Channels (ENaC) are ion channels primarily responsible for sodium reabsorption in epithelial tissues [59]. ENaC in vascular endothelium influences blood pressure by modulating vascular tone. ENaC overexpression in hypertension enhances sodium retention and endothelial stiffness, contributing to atherosclerosis progression [60]. These channels play a crucial role in maintaining salt and water homeostasis, which is essential for regulating blood volume and blood pressure [59]. ENaCs in vascular smooth muscle cells contribute to pressure-induced vasoconstriction [61]. Nav is essential for generating and propagating electrical signals in excitable cells such as neurons, muscles, and cardiomyocytes [62]. Nav channels play a crucial role in cellular excitability by mediating the rapid influx of sodium ions during the upstroke of action potentials [63,64]. Nav1.5 dysfunction in heart failure reduces mitochondrial complex I activity and ATP synthesis. Elevated late sodium current (INa, L) causes Na^+^ accumulation, driving mitochondrial calcium influx via NCX and triggering ROS-induced apoptosis [65].

##### Mg^2+^ Transporters

Mg^2+^ is essential for ATPase activity and mitochondrial oxidative phosphorylation [66]. Dysregulation of Mg^2+^ transporters—particularly TRPM7 and the newly discovered ERMA (TMEM94)—impairs cardiac energetics and promotes CVDs. TRPM7, an Mg^2+^-permeable channel-kinase, maintains intracellular Mg^2+^ homeostasis in cardiomyocytes and vascular smooth muscle cells. Its downregulation in hypertension reduces cytosolic Mg^2+^, leading to calcium overload and vascular stiffness [67]. TMEM94, a P-type ATPase transporter localized to the ER, was recently identified as a key regulator of cardiac Mg^2+^ storage [68]. Disrupted ERMA function causes severe imbalances in cardiac calcium handling, impairing myocardial relaxation and diastolic filling.

### 2.3. Solute Carrier Families

The SLC25 family, also known as the Mitochondrial Carrier Family (MCF), comprises a group of inner mitochondrial membrane proteins essential for the transport of various metabolites, ions, and nucleotides across the mitochondrial membrane [25]. Their activity is regulated by nucleotide binding, calcium ions, and oxidative stress, allowing for dynamic adaptation to cellular energy demands [25]. Adenine nucleotide translocase (ANT), encoded by the *SLC25A4* gene, is a key mitochondrial carrier protein responsible for the exchange of ADP and ATP across the inner mitochondrial membrane. Dysregulation of ANT can disrupt calcium homeostasis, contributing to arrhythmogenic calcium oscillations. This disruption is particularly relevant in arrhythmias, where abnormal calcium signaling can lead to irregular heartbeat [69]. SLC25A10, also known as the dicarboxylate carrier, is a key component of the Malate–Aspartate Shuttle system [25]. It helps maintain mtGSH levels, thereby protecting cells from oxidative stress. *SLC25A1* is located within the chromosome 22 region that is compromised in 22q11.2 deletion syndrome (22q11.2DS), a disorder characterized by congenital heart defects in approximately 60–80% of affected individuals [26]. SLC25A1 has been implicated in metabolic reprogramming and epigenetic regulation in the developing heart. This metabolic dysregulation is associated with impaired TCA cycle flux and oxidative metabolism, leading to an “immature” metabolic profile where the heart remains reliant on glucose for energy production. This metabolic shift may contribute to the observed cardiac malformations by disrupting the normal developmental processes that rely on proper mitochondrial function and metabolic homeostasis. SLC25A20 is a mitochondrial carrier responsible for transporting carnitine and acylcarnitines, which are essential for fatty acid oxidation. This process is crucial for energy production in cardiomyocytes, particularly during periods of high metabolic demand [27]. Dysregulation of SLC25A20 can impair fatty acid oxidation, leading to the accumulation of toxic lipid intermediates and mitochondrial dysfunction.

### 2.4. H^+^ Channels

H^+^ channels, also known as proton channels, are integral membrane proteins that facilitate the movement of hydrogen ions across cell membranes. H^+^ channels are particularly important in excitable cells, such as neurons and cardiomyocytes, where they contribute to membrane potential regulation and electrical signaling [23]. The primary function of Uncoupling Proteins (UCPs) is to regulate mitochondrial proton gradient and ATP synthesis [23,24]. UCP2 can counteract oxidative stress through a “mild uncoupling” process, which reduces ROS production by dissipating the mitochondrial proton gradient. High levels of UCP2 have been shown to reduce inflammation and oxidative stress in endothelial cells, which are key factors in the development of atherosclerosis [70].

In summary, this section systematically classifies mitochondrial transport proteins into distinct functional families—including porin channels (e.g., VDAC isoforms), ion channels (e.g., TRP subfamilies, MCU complex, and K^+^/Na^+^/Mg^2+^ transporters), solute carriers (e.g., SLC25 family and amino acid transporters) and H^+^ channels (e.g., uncoupling proteins)—based on their structural properties, subcellular localization, and physiological roles. It delineates how isoform-specific dysregulation of these transporters (e.g., VDAC1 downregulation impairing fatty acid oxidation in atherosclerosis, TRPC6-mediated Ca^2+^ influx driving cardiac fibrosis, and SLC25A20 deficiency promoting lipotoxicity) directly contributes to cardiovascular pathogenesis by disrupting mitochondrial calcium homeostasis, redox balance, metabolic flexibility, and organelle dynamics. This classification framework, consolidated in Table 1, establishes a foundational link between transporter dysfunction and disease mechanisms, highlighting their potential as precision therapeutic targets across diverse cardiovascular conditions.

## 3. Mitochondrial Transport Proteins/Channels Regulation of CVD Mechanisms

Building on the structural and functional diversity of mitochondrial transporters outlined in Section 2, we now dissect their mechanistic roles in driving cardiovascular pathologies. This section explores how dysregulation of these transport systems—particularly in calcium homeostasis, redox signaling, metabolic reprogramming, inflammation, and vascular function—orchestrates hallmark features of CVDs. We synthesize evidence linking specific transporters (e.g., MCU-induced Ca^2+^ overload, VDAC1-mediated ROS leakage, and TRP channel-driven fibrosis) to disease progression, providing a foundation for targeted therapeutic strategies.

### 3.1. Calcium Homeostasis Regulation

Mitochondrial calcium homeostasis is pivotal for cardiac function, as Ca^2+^ regulates energy production, redox balance, and apoptosis. Dysregulation of mitochondrial Ca^2+^ handling is increasingly linked to CVDs. TRPC1 can interact with other TRPC subunits to form heteromeric channels and is involved in the development of cardiac hypertrophy and heart failure by regulating Ca^2+^-dependent signaling pathways, and also participates in the pathogenesis of arrhythmias by regulating Ca^2+^ signals and ion currents in cardiomyocytes, influencing cardiac electrical activity and rhythm [41,42]. The MCU complex, comprising MCU, MICU1/2, and EMRE, governs mitochondrial Ca^2+^ influx. MCU plays a role in the mediation of mitochondrial calcium uptake, and the occurrence of calcium overload has been demonstrated to activate mPTP and exacerbate ROS production [71,72]. In aging and metabolic stress, reduced MCU activity impairs ATP synthesis, exacerbating cardiac dysfunction [73,74]. In cardiomyocytes, MCU deficiency diminishes contractile reserve, while its overexpression enhances stress tolerance [74]. The mPTP, regulated by cyclophilin D (CypD), opens under Ca^2+^ overload or oxidative stress, triggering mitochondrial swelling and apoptosis. Ca^2+^ overload during reperfusion triggers prolonged mPTP opening via CypD activation, causing mitochondrial swelling and cytochrome c release, directly inducing cardiomyocyte apoptosis in the infarct border zone [75]. Novel approaches, such as nanoparticle-delivered Mg^2+^ and CypD siRNA, synergistically inhibit mPTP overactivation [76], preserving mitochondrial integrity in neurodegenerative models—a strategy potentially adaptable to CVDs.

Furthermore, high-salt diets or chemotherapeutics upregulate TRPC3/6, driving Ca^2+^ overload and pathological remodeling. Genetic or pharmacological inhibition reduces infarct size and fibrosis [19]. TRPC3/6 channels drive pathological calcium overload via biphasic Ca^2+^ gating and an NFATc3-mediated feedforward loop, exacerbating cardiac fibrosis [20,77]. High-salt diets upregulate TRPC6 through angiotensin II signaling, while chemotherapeutics promote TRPC3 membrane translocation via oxidative stress, collectively inducing mitochondrial Ca^2+^ dyshomeostasis and cellular injury [78].

Critically, sarcoplasmic reticulum-mitochondrial microdomain dissociation in heart failure causes oxidative hyperphosphorylation of ryanodine receptors, exacerbating calcium leak and impairing excitation-contraction coupling efficiency. Clinical studies demonstrate a 35% reduction in skeletal muscle calcium handling capacity in heart failure patients, directly correlating with exercise intolerance [79,80]. Furthermore, disrupted calcium oscillations amplify energy crises: while physiological oscillations optimize TCA cycle efficiency via pyruvate dehydrogenase phosphatase activation, sustained pathological calcium signaling uncouples ATP synthesis from oxidative phosphorylation, accelerating cardiomyocyte apoptosis [81].

Mitochondrial calcium overload (>1 μM) directly activates calpain-1 within the matrix, which cleaves complex I subunit NDUFS3 and dynamics protein OPA1, impairing electron transport and inducing mitochondrial fragmentation. This calpain-driven proteolysis exacerbates bioenergetic failure and ROS production, forming a vicious cycle that amplifies mPTP opening and cardiomyocyte death [82,83].

In diabetic cardiomyopathy, UCP2 downregulation elevates ΔΨm, triggering reverse electron transfer (RET)-driven ROS bursts that activate MCU, causing pathological calcium overload. SLC2520 deficiency simultaneously impairs fatty acid oxidation, promoting lipotoxicity that suppresses NCLX-mediated calcium efflux. This dual defect sustains mPTP opening and cardiomyocyte apoptosis [21,84,85]. In Duchenne muscular dystrophy (DMD), sarcolemmal rupture permits uncontrolled calcium influx, overwhelming mitochondrial buffers. The resulting calcium overload directly forces cyclophilin D-dependent mPTP opening, collapsing ATP production and driving necrotic cardiomyopathy through mitochondrial swelling [86,87].

Table 2 summarizes the transporters/complexes involved, their localization, physiological functions, pathological mechanisms associated with CVDs, and potential therapeutic interventions.

### 3.2. Redox Signaling Modulation

Mitochondrial transport proteins critically modulate redox signaling in cardiovascular diseases through compartment-specific regulation of ROS production and scavenging. In atherosclerosis, VDAC1 oligomerization on the mitochondrial outer membrane permeabilizes mitochondria, facilitating ROS leakage and cytochrome c-mediated endothelial apoptosis; this process destabilizes fibrous caps by activating MMPs via mtDNA-cGAS-STING-NLRP3 inflammasome signaling, directly promoting plaque rupture [17,88]. In diabetic cardiomyopathy, TRPA1 overexpression in cardiomyocytes disrupts PPARα-mediated lipid metabolism, causing lipotoxic diastolic dysfunction, whereas its inhibition reverses intramyocardial lipid droplet accumulation by 60% [14,89].

In heart failure, aberrant activation of the HEY2/HDAC1 epigenetic axis is pivotal. Overexpression of the transcriptional repressor HEY2 recruits histone deacetylase HDAC1, compacting chromatin and repressing expression of mitochondrial biogenesis coactivators (e.g., PPARGC1) and fatty acid oxidation enzymes (e.g., CPT1A) [90]. This forces cardiomyocytes to shift from efficient fatty acid oxidation to glycolytic energy production. Concomitant electron transport chain (ETC) dysfunction induces ROS bursts, further damaging mitochondrial DNA and ETC complex stability, thereby establishing a vicious cycle of energy deficit and oxidative damage. In chemotherapy-induced cardiotoxicity, doxorubicin impairs mitochondrial ribosomal protein MRPS5 translation, disrupting ETC complex assembly and reducing superoxide dismutase (SOD2) activity, leading to mtROS accumulation [91]. The E3 ubiquitin ligase TRIM25 counteracts this by degrading the PI3K regulatory subunit p85α, alleviating endoplasmic reticulum stress and suppressing ROS-dependent apoptosis, while the LUZP1 protein activates AMPK phosphorylation to enhance antioxidant gene transcription, restoring redox homeostasis [92,93].

### 3.3. Metabolic Reprogramming

Mitochondrial transporters orchestrate metabolic reprogramming in cardiovascular diseases by dynamically regulating substrate utilization and ion homeostasis, with distinct pathomechanisms in specific CVDs. In atrial fibrillation, MCU deficiency impairs mitochondrial Ca^2+^ uptake, reducing Ca^2+^-dependent activation of pyruvate dehydrogenase (PDH) and isocitrate dehydrogenase. The MCU complex plays a dual role in bioenergetics, while its deficiency impairs Ca^2+^-dependent activation of TCA cycle dehydrogenases, forcing cardiomyocytes to rely on inefficient glycolysis (a Warburg-like effect). Its overactivity under oxidative stress exacerbates Ca^2+^ overload, triggering mPTP opening, ATP depletion, and apoptosis [94,95,96].

Dysregulation of FAO further amplifies metabolic dysfunction. ANT (SLC25A4) dysfunction disrupts ATP/ADP exchange across the inner mitochondrial membrane, reducing oxidative phosphorylation efficiency. In heart failure, ANT1 deficiency impairs mitophagy, leading to accumulation of dysfunctional mitochondria and exacerbating energy deficit. Impaired ANT activity suppresses fatty acid oxidation (FAO) and promotes lipid accumulation in diabetic cardiomyopathy [97], while pharmacological inhibition of carnitine palmitoyltransferase 1 (CPT1) alleviates heart failure but risks lipotoxicity. Furthermore, in DMD, reduced activity of mitoKATP destabilizes ΔΨm, directly compromising mitochondrial calcium buffering capacity [98,99]. Concurrently, amino acid transporters like SLC25A1 and SLC38A2 drive metabolic adaptations—SLC25A1 mutations disrupt citrate export, forcing reliance on glutaminolysis for nucleotide synthesis in hypertrophic cardiomyocytes, whereas SLC38A2 upregulation in heart failure fuels anaplerotic reactions at the cost of ammonia toxicity [100].

### 3.4. Inflammation Participation

Mitochondrial transport proteins critically regulate inflammatory processes in cardiovascular diseases through compartment-specific mechanisms. In atrial fibrillation, TRPM7-mediated Ca^2+^ influx in cardiac fibroblasts activates Ca^2+^/calmodulin-dependent kinase II (CaMKII) and downstream TGF-β1/Smad3 signaling, driving collagen I/III overproduction and fibrotic remodeling. This process is amplified by TRPM7 upregulation (3–5-fold increase in current density) in response to atrial stretch and oxidative stress, directly contributing to electrical conduction heterogeneity in rheumatic heart disease patients [101]. In atherosclerosis, defective ABCA1-mediated cholesterol efflux in macrophages—caused by loss-of-function mutations like p.Glu1282Lys—impairs lipid droplet clearance and promotes NLRP3 inflammasome activation via mitochondrial ROS accumulation. The NLRP3 inflammasome acts as a critical amplifier of vascular inflammation, driving endothelial dysfunction and plaque vulnerability through IL-1β/IL-18 maturation and pyroptosis [102].

In heart failure, MCU complex dysfunction causes mitochondrial calcium overload, which inhibits ETC complex I activity and triggers mtROS bursts; mtROS activates the NF-κB pathway, promoting the release of pro-inflammatory cytokines that accelerate myocardial fibrosis. Concurrently, mtROS further damages ETC complexes, establishing an “energy crisis-oxidative damage-inflammation amplification” vicious cycle [103]. In atherosclerosis, mitochondrial stress in oxLDL-engulfed macrophages induces mtDNA leakage into the cytosol, activating the cGAS-STING pathway to drive type I interferon responses, which promote vascular smooth muscle cell (VSMC) senescence and plaque instability. Notably, nicotinamide mononucleotide (NMN) attenuates plaque development by suppressing aortic inflammation and oxidative stress [104], offering a metabolic intervention strategy. Clinically, decreased mtDNA copy number and elevated ROS in peripheral blood mononuclear cells (PBMCs) of acute ischemic stroke patients negatively correlate with cervicocephalic atherosclerotic burden and predict poor 90-day neurological outcomes [105,106]. In septic cardiomyopathy, lipopolysaccharide downregulates cardiac TMBIM6, releasing its inhibition on VDAC1 oligomerization. This leads to mtDNA release and cGAS-STING activation, culminating in cytokine storms. Myocardial-specific TMBIM6-knockout mice exhibit mitochondrial fission, reduced ATP synthesis, and elevated inflammatory factors—phenotypes reversed by TMBIM6 overexpression or VDAC1 inhibitor NSC15364 [106,107]. Furthermore, in myocarditis, mtDNA amplifies inflammation via the STING pathway. The nanomaterial Nanosweeper targets mtDNA through functional peptides for lysosomal degradation, and combined with STING inhibitor C-176, reduces inflammatory cytokines by 70% in porcine models [108].

### 3.5. Vascular Function Regulation

Mitochondrial homeostasis in VSMCs acts as a central hub in vascular remodeling [109], with dysfunction accelerating CVD progression. TRPC6 can interact with TRPC3 to form signaling complexes, regulating Ca^2+^ influx and ROS production in cardiomyocytes. TRPC6 also contributes to cardiac fibrosis by promoting the proliferation and activation of cardiac fibroblasts and collagen synthesis, affecting cardiac structure and function [110]. TRPV4 activation in vascular smooth muscle cells induces pathological Ca^2+^ influx, triggering mitochondrial ROS overproduction via Ca^2+^-dependent RhoA/ROCK signaling. This promotes vascular stiffness by upregulating collagen deposition and inhibiting elastin synthesis, directly elevating systolic blood pressure in murine models—an effect reversed by TRPV4-selective antagonists like HC-067047 [111]. Beyond hypertension, AP-1-driven transcriptional reprogramming in VSMCs directly triggers aortic dissection and rupture by promoting degenerative phenotypes [112]. Concurrently, Coenzyme Q10 counters FBN1 deficiency-induced VSMC transformation [113], highlighting mitochondrial protectants as modulators of vascular structural integrity.TRPC4 and TRPC1 can regulate vascular endothelial cell proliferation, migration, and NO production by sensing extracellular stimuli and regulating Ca^2+^ signals [114]. Mutations in Nav1.5, the primary Nav channel in cardiomyocytes, are linked to various cardiac arrhythmias, including long QT syndrome and Brugada syndrome [115,116]. Nav channels in vascular smooth muscle cells contribute to pressure-induced vasoconstriction, which is crucial for maintaining vascular tone. Dysregulation of these channels can lead to impaired vascular function and contribute to hypertension [117].

In atherosclerosis, downregulation of the MCU complex reduces mitochondrial matrix calcium, inhibiting pyruvate dehydrogenase activity and the TCA cycle, which causes eNOS uncoupling—decreased NO production and increased superoxide accumulation—thereby exacerbating endothelial dysfunction and monocyte infiltration. Clinically, reduced expression of MICU1 (an MCU regulatory subunit) in human AS plaques negatively correlates with inflammatory severity [118,119]. Concurrently, oscillatory shear stress activates the TRPM2 channel, mediating cytosolic Ca^2+^ influx and NFAT signaling, which drives VSMC transition from a contractile to synthetic phenotype, promoting migration and proliferation and accelerating plaque formation [120,121]. In hypertensive vascular remodeling, mechanical overload activates the Piezo1 channel, inducing mitochondrial Ca^2+^ overload and DRP1-dependent fission, leading to VSMC hyperplasia. Meanwhile, UCP3 downregulation activates the GOT2-aspartate-mTORC1 axis, promoting medial hypertrophy [121,122]. For diabetic vasculopathy, hyperglycemia suppresses the Kir6.1 subunit of the mitoKATP channel, impairing channel opening and exacerbating reverse electron transfer (RET)-dependent ROS bursts during reperfusion, which expands cerebral infarct volume [123].

Collectively, the mechanistic roles of mitochondrial transport proteins drive cardiovascular pathologies through dysregulation of five core processes: calcium homeostasis, redox signaling, metabolic reprogramming, inflammation, and vascular function. Aberrant activity of specific transporters—such as MCU complex-induced Ca^2+^ overload triggering mPTP opening in ischemia–reperfusion injury, VDAC1 oligomerization facilitating ROS leakage in atherosclerosis, and TRPC3/6-mediated Ca^2+^ influx promoting fibrosis—orchestrates hallmark features of CVDs. These disruptions converge on bioenergetic failure, oxidative stress, and maladaptive remodeling, establishing mitochondrial transport machinery as nodal points integrating environmental stressors with disease-specific phenotypes.

Figure 1 provides a comprehensive schematic of these molecular mechanisms, integrating the roles of key transporters (e.g., MCU, VDAC, ANT) in driving CVD progression. This section synthesizes evidence linking transporter dysfunction to clinical outcomes, providing a foundation for targeted interventions.

## 4. Therapeutic Potential of Mitochondrial Transporters in Cardiovascular Diseases

Building upon the established pathogenic roles of mitochondrial transport proteins in CVD progression, this section critically evaluates their emerging therapeutic potential. We synthesize preclinical and clinical advances in targeting dysregulated transporters—including pharmacological modulators, gene-editing technologies, and nanoparticle-based delivery systems—across major cardiovascular conditions.

Figure 2 delineates disease-specific pathogenic cascades mediated by mitochondrial transport dysfunction across major cardiovascular conditions. By bridging mechanistic insights with translational applications, we highlight innovative strategies to rectify transporter-specific dysfunction in heart failure, ischemic injury, cardiomyopathies, atherosclerosis, and myocardial infarction, ultimately proposing a roadmap for precision medicine in mitochondrial-targeted cardioprotection.

### 4.1. Heart Failure

Myocardial dysfunction in heart failure is associated with a reduction in mitochondrial content and oxidative capacity of cardiomyocytes [124]. TASK-3 channels and the mitochondrial NHE, when overexpressed or dysregulated, can exacerbate these issues by promoting mitochondrial dysfunction and cellular apoptosis [79,125]. Impaired ANT function can lead to reduced ATP synthesis and increased oxidative stress. Studies have shown that ANT1-deficient mice exhibit blunted mitophagy and accumulation of aberrant mitochondria, contributing to cardiac dysfunction [126]. Specifically, ANT1 deficiency leads to impaired mitochondrial function and increased oxidative stress. However, recent studies suggest that reducing the rate of fatty acid oxidation may be a beneficial strategy to improve myocardial function [127]. Certain drugs, such as oxfenicine and perhexiline, have been observed to induce the expression of pivotal mitochondrial proteins implicated in cardiac energy metabolism [128]. Through the inhibition of the fatty acid transporter protein CPT1, these medications can potentially retard the progression of heart failure and mitigate detrimental hemodynamic alterations [129]. Furthermore, mPTP formation has been demonstrated to be associated with the process of mitochondrial ATP synthase dimerization [130]. In addition, it has been observed that the administration of cyclosporine A results in the upregulation of SIRT3. The role of SIRT3 in this process involves the inhibition of mPTP through the process of deacetylation of CypD, thereby contributing to the suppression of age-related cardiac hypertrophy [130,131]. In heart failure, calcium overload leads to collapse of mitochondrial membrane potential, excessive opening of mPTP, and triggers apoptosis [132]. Targeting the MCU with small interfering RNA promotes calcium efflux, reduces calcium overload in ischaemia–reperfusion injury, and improves myocardial survival [131].

### 4.2. Ischemic Heart Disease

The Na^+^/Ca^2+^ exchanger is a pivotal component in the pathophysiology of cardiac ischaemia–reperfusion injury [133]. Dysfunction of this exchanger results in calcium overload, which in turn triggers cardiomyocyte apoptosis and necrosis [134]. It has been demonstrated that inhibitors such as SEA0400 and KB-r7943 can mitigate myocardial ischaemia-induced cytoplasmic and mitochondrial calcium overload, thus safeguarding cardiomyocytes [135]. Furthermore, cyclosporin A was found to be cardioprotective, preserving mitochondrial plasma membrane potential and delaying ATP depletion under ischemic conditions [136]. Inhibitors of the MCU, such as Ruthenium red (Ru360), have been shown to reduce infarct size and improve cardiac function in preclinical models [137]. The MCU complex is critical for maintaining normal cardiac rhythm. Studies have demonstrated that inhibiting MCU activity with Ru360 can improve contractile dysfunction and prevent adverse remodeling in models of atrial fibrillation [22]. As of 2025, Ru360 remains in preclinical development due to its poor bioavailability and cellular permeability, dose-dependent toxicity, and complex synthesis requirements [21,138]. Procyclin D inhibitors, such as Debio-025, have been demonstrated to reduce mPTP opening and protect cardiomyocytes [139]. In ischemic heart disease, the SLC25 family is responsible for the exchange of metabolites between the mitochondrial matrix and cytoplasmic retention [140].

In I/R injury, TRPC3-mediated Ca^2+^ influx triggers mPTP opening, mitochondrial swelling, and cardiomyocyte death. Genetic knockout or pharmacological inhibition of TRPC3/6/7 channels reduces infarct size and improves post-ischemic cardiac function [20]. TRPV1 agonists mitigate myocardial I/R injury by preserving mitochondrial membrane potential and ATP synthesis [141]. TRPV1 activation by capsaicin increases mitochondrial membrane potential (MMP) and ATP production in cardiomyocytes subjected to phenylephrine (PE) treatment [141]. This effect is attributed to the promotion of MAM formation and the AMPK-MFN2 pathway [141,142].

### 4.3. Cardiomyopathies

Ca^2+^ channels play a key role in calcium signaling and regulate excitation-contraction coupling in cardiomyocytes, and dysfunction of calcium channels may lead to calcium overload in cardiomyocytes, which in turn may lead to cardiomyopathy [143]. The mitochondrial calcium channels, TRPM2 and TRPA1, are of significant importance in calcium signaling in cardiomyocytes [144]. It has been established that TRPM2 inhibitors, such as ACA, reduce TNF-α-induced calcium overload, thereby attenuating cardiomyocyte death [145]. Conversely, TRPA1 inhibitors, such as HC-030043, have been shown to promote myocardial repair by improving calcium homeostasis [146]. SLC25A49-mediated energy reprogramming governs doxorubicin cardiotoxicity through the G6P–AP-1–Sln axis [147], establishing this transporter as a prime target for gene-editing therapies. T-5224, a specific inhibitor of AP-1, was found to be effective in ameliorating doxorubicin-induced myocardial injury [147]. MCUb overexpression reduces mitochondrial Ca^2+^ uptake conductance by forming low-activity heteromeric channels with MCU. This attenuates pathological cardiac hypertrophy but increases thrombosis risk due to impaired platelet activation [148].

In diabetic cardiomyopathy, hyperglycemia/hyperlipidemia upregulates acid sphingomyelinase to enhance ER-mitochondria contacts, driving MICU1-mediated mitochondrial calcium overload that triggers ROS bursts and apoptosis [85,149]. Concurrent UCP2 downregulation elevates ΔΨm, amplifying complex I reverse electron transfer (RET)-dependent ROS, while SLC25A20 deficiency impairs fatty acid oxidation, suppressing NCLX-mediated calcium extrusion via lipotoxicity, ultimately activating CypD-dependent mPTP opening [150]. In DMD, dystrophin deficiency causes sarcolemmal rupture, inducing Piezo1-dependent Ca^2+^ flooding that overwhelms mitochondrial buffering. Reduced mitoKATP activity destabilizes ΔΨm, directly activating CypD-mPTP, resulting in mitochondrial swelling, ATP depletion, and necrotic cardiomyopathy [99,151].

### 4.4. Coronary Heart Disease

MCU is a core channel for mitochondrial inner membrane-mediated calcium ion uptake, and its activity is regulated by regulatory subunits such as MCUR1, MICU1/2, and EMRE [152]. Abnormal MCU activity in patients with coronary atherosclerotic heart disease leads to mitochondrial calcium overload, triggering ROS burst, impaired ATP synthesis, and apoptosis [153]. The study posits that Ru360 and mitoxantrone function by impeding MCU-mediated calcium inward flow, thus attenuating calcium overload and myocardial injury [154]. VDAC is dysfunctional in coronary atherosclerotic heart disease and is strongly associated with enhanced glycolysis and decreased complex I activity [155]. Bcl-xL inhibitors, such as ABT-737, can block the interaction between VDAC and anti-apoptotic proteins, restoring metabolic balance and alleviating symptoms of coronary atherosclerotic heart disease [18]. ABT-737 remains in the preclinical stage for cardiovascular applications; its clinical translation is limited by unresolved thrombocytopenia risk (due to Bcl-xL inhibition in platelets) and the need for intravenous administration [156]. Chloride channels are involved in mitochondrial membrane potential regulation and ion homeostasis, and their aberrant activation promotes ROS generation and endothelial dysfunction [157]. CLIC1 is highly expressed in atherosclerotic plaques and may contribute to inflammatory responses through the MAPK/ERK pathway [158]. Through virtual screening, the study has found that natural compounds, such as baicalein, can specifically inhibit CLIC1 channel activity and reduce the inflammatory response [158].

### 4.5. Atherosclerosis

Mutations in the ABCA1 gene result in extremely low levels of HDL, as in Tangier disease, which significantly increases the risk of atherosclerosis [159]. ABCG1 knockout mice have decreased HDL levels and increased intraplaque lipid accumulation on a high-fat diet [160]. Clinical trials have shown that the LXR agonist, T0901317, and the PPARγ agonist, rosiglitazone, activate ABCA1/G1 expression, promote cholesterol efflux, increase HDL levels, and reduce plaque volume [159,161]. The restoration of protein function can be achieved through the targeted repair of splice sites or the correction of coding sequence against loss-of-function mutations in ABCA1/G1 or ABCD1 [162]. For example, the dual-guide RNA/SpCas9 system successfully repaired the ABCA4 intron mutation, providing a paradigm for atherosclerosis-related gene therapy [163]. ABCA1 overexpression significantly reduced plaque necrotic core area in animal models [164]. Recombinant adeno-associated virus carrying the ABCA1 gene can target delivery to vascular endothelial cells or macrophages to enhance cholesterol efflux and improve plaque size [164]. ABCD1 is located in the mitochondrial membrane and is involved in the β-oxidation of very long-chain fatty acids (VLCFA), and its deficiency leads to the accumulation of VLCFA, activation of the NLRP3 inflammasome, and release of pro-inflammatory factors such as IL-1β [165]. X-linked adrenoleukodystrophy patients presenting with systemic inflammation and premature atherosclerosis due to ABCD1 mutations [166]. It was shown that phenylbutyric acid acts as a molecular chaperone to facilitate the correct translocation of misfolded ABCD1 proteins to the mitochondrial membrane and restores VLCFA metabolism [166]. In endothelial cells, oxLDL induces VDAC1 upregulation, enhancing cholesterol uptake and mitochondrial ROS generation. This triggers NLRP3 inflammasome activation, promoting endothelial inflammation and atherosclerotic plaque progression. In macrophages, VDAC1-mediated mitochondrial ROS bursts accelerate lipid peroxidation and foam cell formation, key steps in atherogenesis [167]. Endothelial VDAC1 downregulation impairs fatty acid oxidation, leading to lipid accumulation and plaque instability. Recent studies show that G6PD-VDAC1 interaction inhibits Bax-mediated apoptosis, accelerating vascular neointimal hyperplasia—a hallmark of stent restenosis and hypertensive vasculopathy [168]. Resveratrol suppresses VDAC1 overexpression to reduce mPTP opening during myocardial ischemia, confirming its role as a therapeutic node [169]. TRPC1 can regulate vascular endothelial cell proliferation, migration, and NO production by sensing extracellular stimuli and regulating Ca^2+^ signals [170].

### 4.6. Myocardial Infarction

Abnormal opening of mPTP during ischemia–reperfusion leads to mitochondrial swelling, membrane potential collapse, and cytochrome c release, triggering necrosis or apoptosis [171,172]. Cyclophilin D is a core regulator of mPTP, which binds to the adenylate transporter enzyme to promote pore opening. Cyclosporin A inhibits mPTP by binding to Cyclophilin D, which has been shown to reduce infarct size by 20% in clinical trials [173]. Sanglifehrin A, a novel Cyclophilin D inhibitor, is superior to cyclosporin A in preclinical studies for cardio-protection [174]. Upregulated expression of MCU following myocardial infarction has been observed to be associated with calcium homeostasis imbalance [175]. Ru360, which blocks MCU-mediated calcium inward flow, attenuates calcium overload and ROS production, and reduces calcium overload and infarct size in a porcine myocardial infarction model [71,72]. In addition, downregulation of MCU expression using siRNA or CRISPR technology improved calcium homeostasis [176]. Mitochondrial sodium/calcium exchanger activators, such as CGP-37157, also promote calcium efflux, reduce calcium overload, and improve infarct size [71,72].

Taken together, the critical role of mitochondrial transport machinery across diverse cardiovascular pathologies underscores their immense therapeutic potential. To systematically consolidate the emerging pharmacological, genetic, and nanotechnological strategies targeting these proteins, Table 3 provides a comprehensive overview of key mitochondrial targets categorized by cardiovascular disease, and delineates the molecular targets, their subcellular localization, core physiological functions, pathogenic mechanisms in CVDs, and the most promising therapeutic regulators or interventions identified in preclinical and clinical studies. It serves as a strategic roadmap for developing precision therapies that rectify transporter-specific dysfunctions underlying heart failure, ischemic injury, cardiomyopathies, atherosclerosis, and myocardial infarction.

### 4.7. Targeted Delivery Strategies

Targeted delivery of therapeutics to mitochondrial ion channels represents a transformative approach for treating CVDs, yet it faces significant barriers, including poor cellular uptake, lysosomal degradation, and the impermeable double-membrane structure of mitochondria. Recent innovations in nanocarrier design have enabled precise drug delivery to mitochondrial channels [177]. For instance, multistage responsive nanoparticles (e.g., MCTD-NPs) sequentially target ischemic myocardium via EPR effects, escape lysosomes using pH-sensitive polymers, and localize to mitochondria via triphenylphosphonium or SS-31 peptides, enhancing resveratrol delivery to suppress mPTP opening and reduce infarct size by 40% in MI/RI models [178]. Similarly, TPP-conjugated PEG-PE micelles deliver cyclosporin A to inhibit pathological calcium overload through CypD binding, minimizing systemic immunosuppression while improving cardiac function post-MI/RI. Beyond pharmacology, mitochondrial transplantation leverages natural intercellular transfer mechanisms to replenish dysfunctional mitochondria. Clinical studies demonstrate that coronary-infused mitochondria restore endothelial bioenergetics, reduce ROS by 60%, and attenuate vascular inflammation in atherosclerosis via HIF-1α/NF-κB pathway modulation [179,180]. As these strategies advance toward clinical translation, they offer a paradigm shift in treating CVDs rooted in mitochondrial channelopathies.

Collectively, the diverse therapeutic strategies targeting mitochondrial transport proteins across major cardiovascular diseases—ranging from pharmacological modulation and gene editing to nanoparticle delivery and mitochondrial transplantation—are systematically summarized in Figure 3, providing a visual roadmap for future precision interventions.

## 5. Conclusions

### 5.1. Mechanisms of Mitochondrial Transporters

This review identifies mitochondrial transport proteins as central regulators of cardiovascular homeostasis, with distinct isoforms driving CVD pathogenesis through specific mechanisms: (1) calcium homeostasis disruption, mediated by MCU complex hyperactivity (promoting Ca^2+^ overload in myocardial infarction) and TRPC6/3 overactivation (inducing fibrotic remodeling in heart failure); (2) redox imbalance, via VDAC1 oligomerization (facilitating ROS leakage in atherosclerosis) and TRPA1-mediated oxidative stress (exacerbating diabetic cardiomyopathy); (3) metabolic reprogramming, involving ANT dysfunction (impairing ATP/ADP exchange in ischemic heart disease) and SLC25A20 downregulation (reducing fatty acid oxidation in cardiomyopathies); (4) inflammatory activation, such as ABCA1 loss (triggering NLRP3 inflammasome in atherosclerotic plaques) and TRPM7-mediated Ca^2+^ influx (driving cardiac fibrosis).

Dysregulation of key transporters—such as the MCU complex, VDAC isoforms, TRP channels, ANT and SLC25 family members (e.g., SLC25A1 and SLC25A20)—drives bioenergetic failure, oxidative stress, apoptotic signaling, and maladaptive remodeling across CVD spectra, from heart failure and ischemic injury to atherosclerosis and cardiomyopathies. Emerging therapeutic strategies targeting these systems, including pharmacological modulation (e.g., MCU inhibitors like Ru360, mPTP blockers like Cyclosporin A/Sanglifehrin A, TRPV1 agonists like capsaicin, and advanced delivery systems (nanoparticles and gene therapy vectors), and precision gene editing (e.g., CRISPR/Cas9), demonstrate significant preclinical promise in restoring mitochondrial function and alleviating disease progression. These mechanistic insights not only clarify pathogenic pathways but also underscore the translational value of targeting transporters.

### 5.2. Significance of Work

Building on established frameworks of mitochondrial dysfunction in CVDs, we position transport proteins as central arbiters integrating metabolic stress, ionic imbalance, and inflammatory signaling. It positions these proteins not merely as bystanders but as central arbiters integrating metabolic stress, ionic imbalance, oxidative damage, and inflammatory signaling. By comprehensively mapping the structural–functional relationships and isoform-specific pathogenic roles of diverse mitochondrial transporters across the CVD spectrum, this review provides a foundational roadmap for developing mechanistically grounded therapies. It underscores their dual potential as sensitive diagnostic biomarkers and actionable therapeutic targets, offering novel avenues to combat the escalating global burden of CVDs, particularly in aging populations with metabolic comorbidities.

### 5.3. Prospective Perspectives

Based on the above implications, future research should focus on three directions: (1) Multi-targeted calcium modulation, combining MCU inhibitors (e.g., Ru360) with NCLX activators (e.g., efsevin) to synergistically restore Ca^2+^ homeostasis; (2) Isoform-specific drug design, exemplified by VDAC2-selective agonists and TRPC6-specific antagonists that minimize off-target effects; (3) Clinical translation, addressing mitochondrial delivery barriers via stimuli-responsive nanocarriers (e.g., MMP-activated nanoparticles for MCU siRNA). Extending these efforts, multi-omics-guided stratification—using plasma proteomics and lipidomics (e.g., LC-MS-derived ceramide-phosphatidylcholine ratios)—will identify metabolic heart failure subtypes for personalized interventions like SGLT2 inhibitors or malate supplementation. Concurrently, ΔΨm/pH-dual-responsive nanoparticles will exploit pathological mitochondrial hyperpolarization and tissue acidosis to deliver VDAC2 inhibitors (e.g., Cys13-targeted celastrol derivatives) specifically to cardiomyocytes. This approach reduces off-target VDAC1 effects while enhancing anti-apoptotic efficacy (>50% fibrosis reduction in preclinical reperfusion models). Integrating mitochondrial kinase modulators (e.g., AMPK/GSK3β) within these platforms may further preserve ETC function and suppress mPTP opening, unifying metabolic regulation with channel targeting.

## Figures and Tables

**Figure 1 ijms-26-08475-f001:**
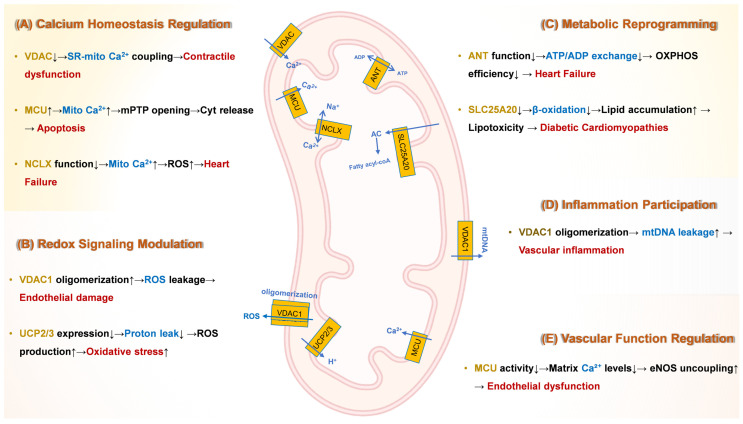
**Molecular Mechanisms Underlying Mitochondrial Transport Protein Dysfunction in Cardiovascular Pathogenesis.** ↓ indicates down-regulation; ↑ indicates up-regulation. Yellow text denotes channel or carrier proteins; blue text indicates key mechanisms; red text signifies downstream effects and consequences. (**A**) Calcium dysregulation: Hyperactivity of the MCU complex induces mitochondrial Ca^2+^ overload, triggering prolonged opening of mPTP. Deficiency in NCLX impairs Ca^2+^ efflux, exacerbating Ca^2+^ overload and oxidative stress in heart failure. and subsequent apoptosis. Dysfunction of VDACs disrupts sarcoplasmic reticulum (SR)-mitochondria Ca^2+^ coupling, leading to impaired excitation-contraction coupling and contractile dysfunction. (**B**) Redox imbalance: Oligomerization of VDAC1 on the OMM increases permeability, facilitating ROS leakage and contributing to endothelial damage. Downregulation of UCP2/3 elevates ΔΨm, promoting reverse electron transfer (RET)-dependent ROS bursts, particularly from complex I. (**C**) Metabolic reprogramming: Dysfunction of ANT reduces ADP/ATP exchange across the IMM, impairing OXPHOS efficiency and contributing to bioenergetic failure in heart failure. Deficiency in SLC25A20 suppresses mitochondrial fatty acid β-oxidation, leading to toxic lipid intermediate accumulation in diabetic cardiomyopathy. Dysregulation of MCU complex shifts cardiac energy metabolism towards less efficient glycolysis, promoting pathological hypertrophy. (**D**) Inflammation: VDAC1 oligomerization facilitates mtDNA leakage into the cytosol. Cytosolic mtDNA activates the cyclic GMP-AMP synthase (cGAS)-stimulator of interferon genes (STING) pathway, driving NLRP3 inflammasome activation, pro-inflammatory cytokine release, and vascular inflammation, destabilizing atherosclerotic plaques. (**E**) Vascular dysfunction: Reduced activity of the MCU complex lowers mitochondrial matrix Ca^2+^ levels. This inhibits Ca^2+^-dependent activation of PDH, suppressing the TCA cycle. ADP, Adenosine diphosphate; ANT, Adenine nucleotide translocator; ATP, Adenosine triphosphate; Ca^2+^, Calcium ion; cGAS, Cyclic GMP-AMP synthase; CypD, Cyclophilin D; ΔΨm, Mitochondrial membrane potential; eNOS, Endothelial nitric oxide synthase; HF, Heart failure; IMM, Inner mitochondrial membrane; MCU, Mitochondrial calcium uniporter; MICU, Mitochondrial calcium uptake; mtDNA, Mitochondrial DNA; mPTP, Mitochondrial permeability transition pore; NCLX, Mitochondrial Na^+^/Ca^2+^ exchanger; NLRP3, NLR family pyrin domain containing 3; NO, Nitric oxide; O_2_^−^, Superoxide; OMM, Outer mitochondrial membrane; OXPHOS, Oxidative phosphorylation; PDH, Pyruvate dehydrogenase; RET, Reverse electron transfer; ROS, Reactive oxygen species; SLC25A20, Solute carrier family 25 member 20 (Carnitine-acylcarnitine translocase); SR, Sarcoplasmic reticulum; STING, Stimulator of interferon genes; TCA, Tricarboxylic acid cycle; UCP, Uncoupling protein; VDAC, Voltage-dependent anion channel.

**Figure 2 ijms-26-08475-f002:**
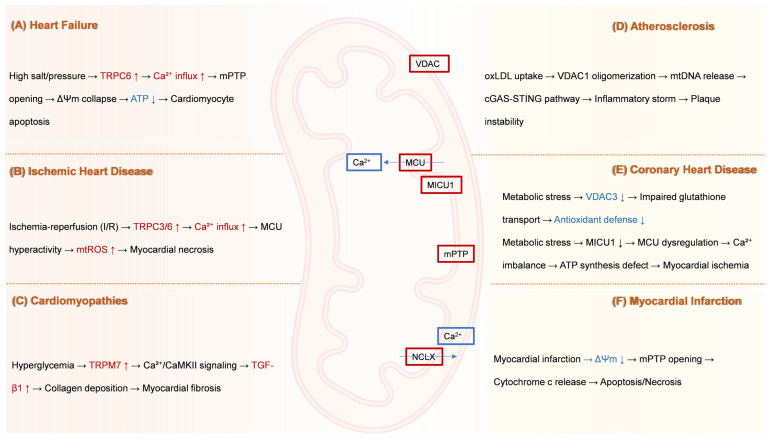
**Disease-specific pathological cascades driven by mitochondrial ion channel and transporter dysfunction.** (**A**) Heart failure: High sodium/pressure stress induces upregulation of TRPC6 channels. This drives pathological Ca^2+^ influx into cardiomyocytes, leading to mPTP opening. mPTP opening collapses ΔΨm, reduces ATP production, and ultimately triggers cardiomyocyte apoptosis. (**B**) Ischemic heart disease: I/R injury causes hyperactivity of TRPC3 and TRPC6 channels. This results in excessive Ca^2+^ influx, inducing MCU hyperactivity. MCU overload elevates miROS production, culminating in myocardial necrosis. (**C**) Cardiomyopathies: Hyperglycemia upregulates TRPM7 channels. Increased Ca^2+^ influx through TRPM7 activates CaMKII signaling. This pathway stimulates TGF-β1 expression, promoting excessive collagen deposition and resulting in myocardial fibrosis. (**D**) Atherosclerosis: Uptake of oxLDL by cells triggers oligomerization of VDAC1 in the mitochondrial outer membrane. VDAC1 oligomerization facilitates mtDNA release into the cytosol. Cytosolic mtDNA activates the cGAS-STING pathway, provoking a robust inflammatory response (“inflammatory storm”) that contributes to atherosclerotic plaque instability. (**E**) Coronary heart disease: Metabolic stress induces two major mitochondrial defects: (1) Downregulation of VDAC3 impairs mitochondrial GSH transport, weakening the organelle’s antioxidant defense system. (2) Downregulation of MICU1 leads to dysregulation of the MCU complex, causing pathological Ca^2+^ imbalance within the mitochondria. This Ca^2+^ dysregulation disrupts ATP synthesis and contributes to myocardial ischemia. (**F**) Myocardial infarction: Acute myocardial infarction causes severe ischemia, leading to a collapse of ΔΨm and consequent mPTP opening. mPTP opening permits the release of cytochrome c from mitochondria into the cytosol, activating pathways of apoptosis and necrosis. Dysfunction of NCLX further exacerbates mitochondrial Ca^2+^ overload, amplifying cell death. ATP, Adenosine triphosphate; Ca^2+^, Calcium ion; CaMKII, Calcium/calmodulin-dependent kinase II; cGAS-STING, cyclic GMP-AMP synthase-stimulator of interferon genes; I/R, Ischemia–reperfusion; MCU, Mitochondrial calcium uniporter; MICU1, Mitochondrial calcium uptake 1; miROS, Mitochondrial reactive oxygen species; mPTP, Mitochondrial permeability transition pore; mtDNA, Mitochondrial DNA; NCLX, Mitochondrial Na^+^/Ca^2+^ exchanger; oxLDL, Oxidized low-density lipoprotein; TGF-β1, Transforming growth factor-beta 1; TRPC, Transient receptor potential canonical; TRPM, Transient receptor potential melastatin; VDAC, Voltage-dependent anion channel; ΔΨm, Mitochondrial membrane potential.

**Figure 3 ijms-26-08475-f003:**
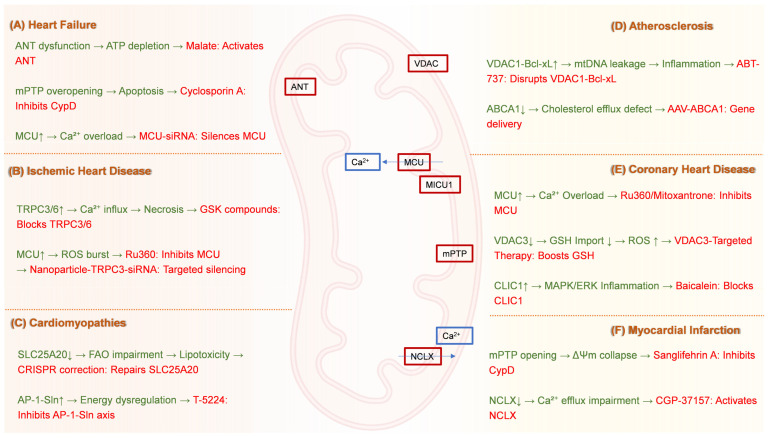
**Mitochondrial Transporter-Targeted Therapeutic Strategies for Cardiovascular Diseases.** (**A**) Heart failure: Cyclosporin A inhibits CypD to suppress mPTP overopening, while MCU-siRNA silences the mitochondrial calcium uniporter to prevent Ca^2+^ overload. (**B**) Ischemic heart disease: GSK compounds block TRPC3/6 channels to reduce pathological Ca^2+^ influx; Ru360 inhibits MCU to attenuate ROS generation; Nanoparticle-TRPC3-siRNA enables targeted channel silencing. (**C**) Cardiomyopathies: CRISPR correction repairs SLC25A20 mutations to restore fatty acid oxidation, and T-5224 inhibits the AP-1-Sin axis to counteract energy dysregulation. (**D**) Atherosclerosis: ABT-737 disrupts VDAC1–Bcl-xL interactions to prevent mtDNA leakage, while AAV-ABCA1 gene therapy enhances cholesterol efflux via ATP-binding cassette transporter A1 delivery. (**E**) Coronary heart disease: Ru360 or Mitoxantrone inhibits MCU to alleviate Ca^2+^ overload; VDAC3-targeted therapy boosts GSH import to reduce ROS; Baicalein blocks CLIC1 to suppress MAPK/ERK-driven inflammation. (**F**) Myocardial infarction: Sanglifehrin A inhibits CypD to prevent mPTP opening, and CGP-37157 activates the NCLX to restore Ca^2+^ efflux. CypD, Cyclophilin D; mPTP, Mitochondrial permeability transition pore; MCU, Mitochondrial calcium uniporter; Ca^2+^, Calcium ion; siRNA, Small interfering RNA; TRPC, Transient receptor potential canonical; ROS, Reactive oxygen species; CRISPR, Clustered regularly interspaced short palindromic repeats; SLC25A20, Solute carrier family 25 member 20; FAO, Fatty acid oxidation; AP-1-Sin, AP-1-Sin3 transcriptional regulatory axis; VDAC, Voltage-dependent anion channel; Bcl-xL, B-cell lymphoma-extra large; mtDNA, Mitochondrial DNA; AAV, Adeno-associated virus; ABCA1, ATP-binding cassette transporter A1; GSH, Glutathione; CLIC1, Chloride intracellular channel 1; MAPK/ERK, Mitogen-activated protein kinase/extracellular signal-regulated kinase; NCLX, Mitochondrial Na^+^/Ca^2+^ exchanger.

**Table 1 ijms-26-08475-t001:** Classification of mitochondrial transport proteins and their roles in cardiovascular diseases.

Family	Key Members	Localization	Core Physiological Functions	Roles in Cardiovascular Diseases	Therapeutic Targets/Strategies	Refs.
Anion channels	VDAC1/2/3	OMM	ATP/ADP transport; Ca^2+^ homeostasis	VDAC1 oligomerization → apoptosis (HF, IHD); VDAC3 →antioxidant defense (atherosclerosis)	VDAC1 inhibitors (e.g., ABT-737); VDAC3 targeting	[17,18]
TRP channels	TRPC3	IMM	Ca^2+^/Na^+^ influx; oxidative stress regulation	TRPC3 overexpression → Ca^2+^ overload → fibrosis (hypertension, HF);	TRPC3 inhibitors;	[19,20]
Metal ion transporters	MCU Complex (MCU, MICU1/2)	IMM	Ca^2+^ influx; ATP synthesis	MCU overactivation → Ca^2+^ overload → mPTP opening → apoptosis (MI, I/R injury); MICU1/2 deficiency → HF	MCU inhibitors (e.g., Ru360); NCLX activators	[21,22]
H^+^ channels	UCP2/3	IMM	Proton leak; ROS reduction; energy metabolism	UCP2↓→ endothelial oxidative stress/inflammation (atherosclerosis); UCP3↑→ cardioprotection (I/R)	UCP2/3 activators	[23,24]
SLC25 Family (Metabolite transporters)	ANT (SLC25A4), SLC25A20	IMM	ATP/ADP exchange; FAO promotion	ANT dysfunction → ATP synthesis↓ (HF); SLC25A20 deficiency → lipid accumulation → mitochondrial dysfunction (diabetic cardiomyopathy)	ANT activators; SLC25A20 targeting	[25,26,27]

↓ indicates down-regulation; ↑ indicates up-regulation. VDAC, Voltage-dependent anion channel; OMM, Outer mitochondrial membrane; HF, Heart failure; IHD, Ischemic heart disease; TRP, Transient receptor potential; IMM, Inner mitochondrial membrane; MCU, Mitochondrial calcium uniporter; mPTP, Mitochondrial permeability transition pore; MI, Myocardial infarction; I/R, Ischemia–reperfusion; NCLX, Mitochondrial Na^+^/Ca^2+^ exchanger; ER, Endoplasmic reticulum; OXPHOS, Oxidative phosphorylation; ANT, Adenine nucleotide translocator; SLC25A20, Solute carrier family 25 member 20 (carnitine-acylcarnitine translocase); FAO, Fatty acid oxidation; UCP, Uncoupling protein; ROS, Reactive oxygen species.

**Table 2 ijms-26-08475-t002:** Key transporters in mitochondrial calcium homeostasis and their roles in cardiovascular pathogenesis.

Transporters/Complexes	Localization	Physiological Functions	Pathological Mechanisms	Therapeutic Interventions	Refs.
MCU Complex	IMM	Mediates high-calcium Ca^2+^ influx (>1 μM); activates TCA cycle enzymes; promotes ATP production	Over-activation → mitochondrial Ca^2+^ overload → mPTP opening → apoptosis (MI, I/R injury); Reduced activity → decreased ATP synthesis	Ru360, mitoxantrone (inhibitors); Oleuropein (activator)	[72,74]
NCLX	IMM	Mediates Ca^2+^ efflux (Na^+^ exchange); balances MCU-mediated influx; maintains Ca^2+^ homeostasis	Deficiency → impaired Ca^2+^ efflux → exacerbates Ca^2+^ overload/oxidative stress (I/R injury)	Efsevin (activator)	[84]
VDAC2/3	OMM	VDAC2 stabilizes Ca^2+^ uptake; VDAC3 enables high Ca^2+^ permeability; regulates ER-mitochondria Ca^2+^ signaling	VDAC2 deletion → Ca^2+^ imbalance → dilated cardiomyopathy; VDAC1 oligomerization → enhanced Ca^2+^ leakage → apoptosis (HF, MI)	VDAC2 targeting (enhances buffering); VDAC1 oligomerization inhibitors (e.g., compound C)	[32]
TRPC3	IMM	Mediates Ca^2+^ influx; participates in oxidative stress/cell proliferation signaling	High-salt/chemotherapy-induced overexpression → Ca^2+^ overload → myocardial fibrosis (hypertension, chemo-induced cardiomyopathy)	Genetic knockout/pharmacological inhibition (e.g., HC-030043)	[20,77,78]
mitoBKCa Channel	IMM	Calcium-activated K^+^ efflux; stabilizes ΔΨm; prevents Ca^2+^ overload	Over-activation → ΔΨm collapse → impaired OXPHOS → increased oxidative stress (HF)	Channel activity regulation	[53]

MCU, Mitochondrial calcium uniporter; IMM, Inner mitochondrial membrane; Ca^2+^, Calcium ion; mPTP, Mitochondrial permeability transition pore; MI, Myocardial infarction; I/R, Ischemia–reperfusion; TCA, Tricarboxylic acid cycle; ATP, Adenosine triphosphate; NCLX, Mitochondrial Na^+^/Ca^2+^ exchanger; VDAC, Voltage-dependent anion channel; OMM, Outer mitochondrial membrane; HF, Heart failure; TRPC, Transient receptor potential canonical; mitoBKCa, Mitochondrial big-conductance Ca^2+^-activated K^+^ channel; ΔΨm, Mitochondrial membrane potential.

**Table 3 ijms-26-08475-t003:** Mitochondrial transporters as therapeutic targets in cardiovascular diseases.

Disease	Targeted Mitochondrial Transporter	Pathogenic Role	Therapeutic Strategy	Mechanism of Intervention	Refs.
Heart failure	ANT	Impaired ADP/ATP exchange leads to energetic deficit	Malate supplementation	Activates ANT to restore ATP synthesis	[126,127]
	MCU Complex	Ca^2+^ overload	MCU-siRNA	Suppresses MCU-mediated Ca^2+^ influx	[131]
	mPTP	Permeability transition leads to ΔΨm collapse	Cyclosporin A	Binds CypD to inhibit mPTP opening	[130]
Ischemic heart disease	TRPC3/6	Pathological Ca^2+^ influx	TRPC3/6 inhibitors (GSK compounds)	Blocks Ca^2+^ entry channels	[20]
	MCU Complex	Mitochondrial Ca^2+^ uptake	Ru360	Directly inhibits MCU pore	[22]
	ANT	ATP depletion	Energy crisis from ATP depletion	Enhances ANT-driven ADP/ATP exchange	[98]
Cardiomyopathies	TRPM2	Ca^2+^ overload leads to cardiomyocyte death	ACA	Inhibits TRPM2 channel activity	[145]
	SLC25A49	Metabolic reprogramming and doxorubicin toxicity	T-5224 (AP-1 inhibitor)	Suppresses AP-1-mediated Sln overexpression	[147]
Atherosclerosis	ABCA1/ABCG1	Defective cholesterol efflux contributes to foam cells	T0901317 (LXR agonist)	Upregulates ABCA1/G1 expression	[102]
	VDAC1	NLRP3 inflammasome activation	ABT-737	Disrupts VDAC1-Bcl-xL interaction to inhibit apoptosis	[169]
Coronary heart disease	MCU complex	Ca^2+^ homeostasis	Ru360/Mitoxantrone	Inhibits MCU-mediated Ca^2+^ uptake	[153,154]
	VDAC	Enhanced glycolysis & reduced complex I activity	VDAC3-targeted therapy	Boosts glutathione import to combat oxidative stress	[106]
Myocardial infarction	mPTP/CypD	I/R-induced mitochondrial swelling	I/R-induced mitochondrial swelling	Sanglifehrin A (CypD inhibitor)	[171]
	MCU	Ca^2+^ efflux	CRISPR-MCU editing	Gene-editing knockdown of MCU expression	[71,72]

ANT, Adenine nucleotide translocator; ADP, Adenosine diphosphate; ATP, Adenosine triphosphate; MCU, Mitochondrial calcium uniporter; siRNA, Small interfering RNA; mPTP, Mitochondrial permeability transition pore; CypD, Cyclophilin D; ΔΨm, Mitochondrial membrane potential; TRPC, Transient receptor potential canonical; OXPHOS, Oxidative phosphorylation; TRPM, Transient receptor potential melastatin; SLC25A49, Solute carrier family 25 member 49; AP-1, Activator protein 1; Sln, Sarcolipin; ABCA1/ABCG1, ATP-binding cassette subfamily A/G member 1; LXR, Liver X receptor; NLRP3, NLR family pyrin domain containing 3; VDAC1, Voltage-dependent anion channel 1; Bcl-xL, B-cell lymphoma-extra large; TRPV1, Transient receptor potential vanilloid 1; FAO, Fatty acid oxidation; I/R, Ischemia–reperfusion; NCLX, Mitochondrial Na^+^/Ca^2+^ exchanger.

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
