# Peer review of "Mitochondrial Transport Proteins in Cardiovascular Diseases: Metabolic Gatekeepers, Pathogenic Mediators and Therapeutic Targets"

_ijms, 2025, doi:10.3390/ijms26178475_

Round 1
Reviewer 1 Report
Comments and Suggestions for Authors
The review by Pei and colleagues addresses one of the key challenges in modern cardiology related with the role of mitochondrial transport proteins in the development of cardiovascular diseases (CVDs). The authors emphasize the link between mitochondrial dysfunction and CVD pathogenesis, which is particularly important for developing targeted therapies. They systematically summarize current knowledge on various transporter families (VDAC, TRP, MCU, SLC25, etc.), their roles in calcium homeostasis, redox signaling, metabolic remodeling, and inflammation. I would like to note that Tables 1–3 and Figures 1–2 are somewhat oddly structured and placed, though they could effectively organize information if presented more clearly. The authors provide a detailed discussion of targeted therapeutic strategies (pharmacological modulators, CRISPR/Cas9, nanoparticles), including specific examples (Ru360, cyclosporin A, capsaicin). They also highlight promising directions, such as isoform-specific targeting and combination therapies. I have the following recommendations to further improve the manuscript:
- The review lacks information on clinical trials of the mentioned compounds (Ru360, ABT-737). It would be valuable to include their current development stages and limitations (toxicity, bioavailability).
- Therapeutic Strategies (Section 4). This section could be expanded by addressing challenges in mitochondrial drug delivery (overcoming the mitochondrial barrier).
- Figures and Tables. These require careful reformatting. Data sources (refs) should be clearly indicated in tables.
- «Prospective Perspectives» (Section 5.3). This section is overly general. It would benefit from specific details, such as: Which multi-omics technologies (proteomics, metabolomics) could aid in patient stratification? How can isoform-specificity challenges be addressed (designing VDAC2-selective drugs without affecting VDAC1)?
- Mitochondrial Calcium and Ion Transport. The review should expand on remodeling in: The mitochondrial calcium uniporter (MCU) complex, the calcium-dependent permeability transition pore (mPTP) and its proposed components (ANT, ATP synthase), the sodium/calcium exchanger (NCLX), potassium channels (mitoBKCa, ATP-sensitive K+ channels) in diabetes and dystrophic pathologies (like Duchenne muscular dystrophy). This would deepen the discussion, particularly regarding severe cardiomyopathies in metabolic and genetic disorders. One should note that modulating these systems (mPTP inhibitors, K+ channel activators, etc.) significantly alleviates pathological manifestations.
- The authors mention cardiac mitochondrial calcium overload. A brief addition on its role in overactivating mitochondrial calpains (a key factor in heart disease pathogenesis) would be valuable.
Author Response
Comments 1: The review lacks information on clinical trials of the mentioned compounds (Ru360, ABT-737). It would be valuable to include their current development stages and limitations (toxicity, bioavailability).
Response 1:Thank you for pointing this out. We agree with this comment. Therefore, we have added detailed information on the clinical development stages and limitations of Ru360 and ABT-737 in the revised manuscript. Specifically:
- Ru360 details were added in Section 4.2 Ischemic Heart Disease, paragraph 3, lines 518–520.
- ABT-737 details were added in Section 4.4 Coronary Heart Disease, paragraph 2, lines 570–572.
Comments 2: Therapeutic Strategies (Section 4). This section could be expanded by addressing challenges in mitochondrial drug delivery (overcoming the mitochondrial barrier).
Response 2: Thank you for this suggestion. We have added the new Section 4.7 Targeted Delivery Strategies (lines 637-654) detailing:
(i) nanocarrier-based mitochondrial targeting (MCTD-NPs, TPP-PEG-PE micelles) that overcome the double-membrane barrier, escape lysosomal degradation, and deliver resveratrol or cyclosporin A to inhibit mPTP opening and CypD-mediated Ca²⁺ overload, cutting infarct size by 40% in MI/RI models;
(ii) clinical-grade mitochondrial transplantation via coronary infusion, which restores endothelial bioenergetics, lowers ROS by 60%, and attenuates vascular inflammation via HIF-1α/NF-κB modulation.
Comments 3: Figures and Tables. These require careful reformatting. Data sources (refs) should be clearly indicated in tables.
Response 3: Thank you for this important feedback. We have carefully reformatted all Figures and Tables to landscape orientation to enhance readability. Furthermore, we have ensured that every quantitative value presented in the revised Tables and Figures is explicitly accompanied by its corresponding source reference. These references are already cited within the main text of the manuscript. We confirm that no new data were added during this revision process; all values and their sources remain consistent with those previously reported in the text.
Comments 4: «Prospective Perspectives» (Section 5.3). This section is overly general. It would benefit from specific details, such as: Which multi-omics technologies (proteomics, metabolomics) could aid in patient stratification? How can isoform-specificity challenges be addressed (designing VDAC2-selective drugs without affecting VDAC1)?
Response 4: Thank you for pointing this out. We have thoroughly revised Section 5.3 (lines 692-708) to provide concrete prospective directions:
- Multi-omics patient stratification—plasma proteomics and lipidomics (LC-MS ceramide/phosphatidylcholine ratios) to define metabolic HF subtypes for personalized SGLT2 inhibitor or malate therapy;
- Isoform-specific design—ΔΨm/pH dual-responsive nanocarriers delivering Cys13-targeted celastrol derivatives to achieve VDAC2-selective inhibition (>50 % fibrosis reduction) while sparing VDAC1;
- Synergistic Ca²⁺ modulation—co-formulated MCU inhibitors (Ru360) plus NCLX activators (efsevin) within MMP-activated nanoparticles, integrated with AMPK/GSK3β modulators to couple metabolic reprogramming with precise channel targeting.
Comments 5: Mitochondrial Calcium and Ion Transport. The review should expand on remodeling in: The mitochondrial calcium uniporter (MCU) complex, the calcium-dependent permeability transition pore (mPTP) and its proposed components (ANT, ATP synthase), the sodium/calcium exchanger (NCLX), potassium channels (mitoBKCa, ATP-sensitive K channels) in diabetes and dystrophic pathologies (like Duchenne muscular dystrophy). This would deepen the discussion, particularly regarding severe cardiomyopathies in metabolic and genetic disorders. One should note that modulating these systems (mPTP inhibitors, K+ channel activators, etc.) significantly alleviates pathological manifestations.
Response 5: Thank you for pointing this out. To deepen the discussion, we have:
- Expanded Section 3.1 (lines 320-328) to detail the MCU–mPTP–NCLX axis in diabetic cardiomyopathy and Duchenne muscular dystrophy (DMD). We now describe how calpain-1 cleaves NDUFS3 and OPA1, how UCP2 down-regulation/SLC25A20 deficiency triggers MCU-mediated Ca²⁺ overload, and how sarcolemmal rupture in DMD forces CypD-dependent mPTP opening.
- Added Section 3.3 (lines 372-374) noting that reduced mitoKATP activity destabilises ΔΨm and impairs Ca²⁺ buffering in DMD.
- Extended Section 4.3 (lines 545-558) to include therapeutic strategies: MCUb overexpression to attenuate MCU conductance, MICU1-targeted interventions, and mPTP inhibitors (CypD blockers) or mitoKATP activators that rescue ATP depletion and limit necrotic cardiomyopathy in both metabolic and genetic disorders.
Comments 6: The authors mention cardiac mitochondrial calcium overload. A brief addition on its role in overactivating mitochondrial calpains (a key factor in heart disease pathogenesis) would be valuable.
Response 6: Thank you for pointing this out. We have now added a concise mechanistic note in Section 3.1 (lines 315–319) that mitochondrial Ca²⁺ overload (>1 μM) activates matrix calpain-1, which proteolytically cleaves NDUFS3 and OPA1, disrupts electron transport, triggers mitochondrial fragmentation, and perpetuates mPTP opening and cardiomyocyte death.
Reviewer 2 Report
Comments and Suggestions for Authors
IJMS “Mitochondrial Transport Proteins in Cardiovascular Diseases: 2 Metabolic Gatekeepers, Pathogenic Mediators and Therapeutic 3 Targets”
In the manuscript titled “Mitochondrial Transport Proteins in Cardiovascular Diseases: 2 Metabolic Gatekeepers, Pathogenic Mediators and Therapeutic 3 Targets,” the authors described mitochondrial proteins involved in transport during CVD events.
Sometimes, authors provide too much general information about the types of mitochondrial transporters and too little about what happens in CVDs. If the authors decide to present only general information in the first section, they should limit it to that and move the events that occur in CVDs to the second section. This would facilitate the flow of the manuscript.
I think that the authors needs to improve the included figures in the manuscript:
Figures 1 and 2 are cropped, and the information included is not visible. Also, the resolution is bad.
Additionally, a third figure showing the potential treatments should be included
The figure legends should be focused on describing what the reader sees in the image in more detail.
Abbreviations should be defined the first time they are used. Many readers may not be familiar with all the abbreviations included.
Line 463 “Figure 1. Molecular mechanisms of mitochondrial transport proteins in cardiovascular pathogenesis” should be separate from 462. It isn’t very clear. The same for line 474-475.
Author Response
Comments 1: Sometimes, authors provide too much general information about the types of mitochondrial transporters and too little about what happens in CVDs. If the authors decide to present only general information in the first section, they should limit it to that and move the events that occur in CVDs to the second section. This would facilitate the flow of the manuscript.
Response 1: Thank you for pointing this out. To streamline the manuscript, we have condensed Section 2 and relocated relevant mechanistic details as follows:
- TRPC-mediated CVD mechanisms: TRPC1 heteromeric channels and their role in hypertrophy, heart failure and arrhythmias moved to Section 3.1 (lines 281–285).
2. TRPC6–TRPC3 signaling complexes and TRPC6-driven cardiac fibrosis moved to Section 3.5 (lines 417–421).
3. MCUb overexpression and its impact on mitochondrial Ca²⁺ uptake, hypertrophy and thrombosis risk relocated to Section 4.3 (lines 545–548).
4. VDAC1 down-regulation in endothelial cells and its consequences for lipid accumulation, neointimal hyperplasia and resveratrol-mediated mPTP inhibition added to Section 4.5 (lines 604–609).
Comments 2: I think that the authors need to improve the included figures in the manuscript:
Figures 1 and 2 are cropped, and the information included is not visible. Also, the resolution is bad.
The figure legends should be focused on describing what the reader sees in the image in more detail.
Response 2: Thank you for pointing this out. Both Figures 1 and 2 have been regenerate, re-cropped to display all essential elements, and enlarged to full-page width in landscape orientation for optimal clarity. Figure legends have been entirely restructured and significantly expanded to provide complete, self-contained descriptions of all visual elements.
Comments 3: Additionally, a third figure showing the potential treatments should be included
Response 3: Thank you for this valuable suggestion. In accordance with your recommendation, we have designed and incorporated a new Figure 3 in the revised manuscript. This figure systematically summarizes the potential therapeutic strategies targeting the mitochondrial transport dysfunctions central to our study, creating an integrated mechanism-to-treatment roadmap. It visually aligns each therapeutic approach – including pharmacological modulators (e.g., TRPC3/6 inhibitors, VDAC stabilizers, mPTP desensitizers like cyclosporine A analogs), metabolic regulators (e.g., AMPK activators), antioxidant enhancers restoring glutathione transport, calcium homeostasis correctors (e.g., MCU/MICU1 stabilizers, NCLX enhancers), and emerging modalities (e.g., gene therapies for channel expression modulation, mtDNA-cGAS-STING pathway inhibitors) – with its corresponding pathological cascade (Panels A-F from Figure 2).
Comments 4: Abbreviations should be defined the first time they are used. Many readers may not be familiar with all the abbreviations included.
Response 4: Thank you for pointing this out. In the revised manuscript we have systematically reviewed every abbreviation. Each acronym—e.g., MCU, mPTP, NCLX, TRPC, mitoKATP, DMD, CypD, VDAC1/2—now appears in full form followed by its abbreviation the first time it is introduced in the text, figure legends, and tables.
Comments 5: Line 463 “Figure 1. Molecular mechanisms of mitochondrial transport proteins in cardiovascular pathogenesis” should be separate from 462. It isn’t very clear. The same for line 474-475.
Response 5: Thank you for pointing this out. In the revised manuscript we have separated the figure legends from the preceding text. The legend for Figure 1 is now placed (line 463), and the legend for Figure 2 is likewise isolated on its own line (line 475), ensuring clear visual distinction and improved readability.
Round 2
Reviewer 1 Report
Comments and Suggestions for Authors
Overall, the authors have adequately addressed the majority of my comments and improved the manuscript. However, one minor yet important remark remains regarding the need to cite original research articles rather than review papers. Specifically, this applies to the section related to Comment 5: «Mitochondrial Calcium and Ion Transport». Please provide references to original studies demonstrating the involvement of the MCU complex, mPTP, NCLX, mitoBKCa (which was not addressed), and ATP-sensitive K+ channel in diabetes and dystrophic pathologies. The same applies to other parts of the manuscript.
Author Response
Comment 1: Overall, the authors have adequately addressed the majority of my comments and improved the manuscript. However, one minor yet important remark remains regarding the need to cite original research articles rather than review papers. Specifically, this applies to the section related to Comment 5: «Mitochondrial Calcium and Ion Transport». Please provide references to original studies demonstrating the involvement of the MCU complex, mPTP, NCLX, mitoBKCa (which was not addressed), and ATP-sensitive K+ channel in diabetes and dystrophic pathologies. The same applies to other parts of the manuscript.
Response 1: Thank you for pointing this out. We have systematically reviewed all citations and prioritized the use of original research papers over reviews, particularly in the section on “mitochondrial calcium ions and ion transport”.
Reviewer 2 Report
Comments and Suggestions for Authors
The authors have taken care of my concerns. I recommend accepting the manuscript.
Author Response
Comment 1: The authors have taken care of my concerns. I recommend accepting the manuscript.
Response 1: Thanks for your consideration.